# Linguistic Compression in Single-Sentence Human-Written Summaries

**Fangcong Yin**
Department of Information Science
Cornell University
fy95@cornell.edu

**Marten van Schijndel**
Department of Linguistics
Cornell University
mv443@cornell.edu

## Abstract

Summarizing texts involves significant cognitive efforts to compress information. While advances in automatic summarization systems have drawn attention from the NLP and linguistics communities to this topic, there is a lack of computational studies of linguistic patterns in human-written summaries. This work presents a large-scale corpus study of human-written single-sentence summaries. We analyzed the linguistic compression patterns from source documents to summaries at different granularities, and we found that summaries are generally written with morphological expansion, increased lexical diversity, and similar positional arrangements of specific words compared to the source across different genres. We also studied how linguistic compressions of different factors affect reader judgments of quality through a human study, with the results showing that the use of morphological and syntactic changes by summary writers matches reader preferences while lexical diversity and word specificity preferences are not aligned between summary writers and readers.

## 1 Introduction

Summarization requires condensing important information from a longer text into a brief form (Johnson, 1983). It involves complex cognitive capabilities related to logical thinking and language proficiency and has been commonly used in classroom settings to teach and assess these capabilities of students. Extensive education and psychology research on the development of students has shown that summary writing is a rich resource to study the human cognitive process of comprehension, retention, and organization (Brown et al., 1983).

Recently, human summary writing has drawn increased attention from the NLP and psycholinguistics communities for the purposes of computational modeling of language production. Given advances in transformer-based neural models, automatic text summarization systems have achieved a breakthrough in performance. But abstractive summaries generated by neural summarizers still suffer from hallucinated generation (Maynez et al., 2020; Wang et al., 2022) and verbose copying from the source text (Wilber et al., 2021; Zhang et al., 2023). Studies have shown that mimicking human behaviors and preferences is a promising approach toward more naturalistic generations for machine-learning tasks (Jaidka et al., 2013; Xiao et al., 2022) and consistent patterns of human summary writing can provide helpful references to this end.

Previous literature has investigated cognitive operations behind human summary writing, especially the operation of compression. Compressing text into a concise form requires the deletion of irrelevant information, generalization of concepts, and construction of fluent expressions (Hidi and Anderson, 1986). This entails complicated operations on the language features of the produced texts at the levels of syntax, semantics, and discourse. The strategies of compression during summarization can be studied using corpus linguistics (Sherrard, 1989). However, there is a lack of fine-grained analysis of the compression strategies present in summarization, and a dearth of large-scale quantitative studies that examine the general patterns of linguistic compression strategies in human summary writing.

To provide more in-depth insight into human behaviors during summary writing, this study presents a large-scale computational analysis of linguistic compression patterns in human-written summaries of expository texts. To amplify any consistent linguistic compression patterns, we focus on *extreme summaries* (Narayan et al., 2018), i.e. single-sentence summaries that are highly compressed and abstractive without verbatim copying from the source text. The patterns are extracted from large summarization corpora of different sub-

| | # Source-Summary Pair (train/val/test) | Avg Source Length (sd) | Avg Summary Length (sd) |
|---|---|---|---|
| **WikiHow** | 1060732/37932/41182 | 82.32 (60.10) | 8.28 (5.27) |
| **XSum** | 204045/11332/11334 | 433.05 (355.51) | 23.19 (5.81) |
| **SciTLDR-Auth** | 1992/618/618 | 1142.09 (469.71) | 20.34 (8.38) |
| **SciTLDR-Peer** | 1992/834/1349 | 1142.09 (469.71) | 23.17 (7.65) |

Table 1: Descriptive statistics of the datasets used in this paper (in tokens).

genres at multiple granularities to obtain a more fine-grained picture of human summary writing. We then conduct a human study to evaluate which linguistic compressions are most preferred by readers.

## 2 Related Work

The cognitive operations of compression during summary writing were formulated as "macrorules" by Brown and Day (1983). Brown and Day proposed four common operations in summary writing as macrorules, which experienced high school and college students self-reported to intentionally and frequently use when writing summaries: deletion, selection, substitution, invention, and combination. Empirical research has shown that linguistic features can reflect the usage of some of the macrorules (Sherrard, 1989). Denhiere (2005) proposed a computational model using sentence representations and latent semantic analysis to model the use of macrorules by participants in an experimental study.

Recently, studies in journalism and education have investigated linguistic features of summaries for specific domains by viewing news headlines (Piotrkowicz et al., 2017; Xu and Guo, 2018), research article abstracts (Amnuai et al., 2020), and paper reviews (Leijen and Leontjeva, 2012) as summaries. These works have identified patterns for each domain and shown that the patterns can help writers improve the quality and effectiveness of their summary writing. For example, Leijen and Leontjeva (2012) examined the correlations between the linguistic features of peer reviews of academic papers and the acceptability of the revision suggestions provided in the reviews using a random forest model, and found that the paper authors are more likely to accept and implement the reviewer comments if the comments have more directives. Piotrkowicz et al. (2017) studied news headlines as a form of extreme summaries and correlated

the usage of sentiment, named entities, phrases, syntactic structure, and punctuation in news headlines with the popularity of the associated articles. The only cross-domain study of the linguistic influences on the general task of summary writing is Arroyo-Fernández et al. (2019), which compared the differences in the use of several language features, including part-of-speech tags, sentence sentiment, named entity tags, and relation tags from rhetoric structure theory, between human-written and machine-generated summaries. Arroyo-Fernández et al. found that machine-generated summaries show significantly different behaviors in the usage of named entities from humans. However, these existing works did not discuss the fine-grained distributional differences between source document and summary pairs or evaluate how the identified linguistic patterns are related to the perceived summary quality by summary readers. In contrast, we propose a more fine-grained analysis framework of the linguistic patterns present in summarization, analyzing the pairwise relations between source documents and the associated summaries at different granularities in addition to linguistic features of the summaries in isolation.

## 3 Linguistic Compression Patterns

### 3.1 Data

We focused on extreme summarization of expository texts, the task of creating a single-sentence abstractive summary of a single document (Narayan et al., 2018). Extreme summarization is a challenging yet commonly studied task by the automatic text summarization community. Given its tight length constraint, extreme summarization requires more abstractive and compressed summaries compared with other forms of summarization, helping ensure that non-trivial efforts of compression will occur in summarization. In addition, extreme summary datasets tend to have fewer supplemental materials that are non-essential in the source docu-

ment (Bommasani and Cardie, 2020), providing a purer window on the source compression.

We selected three popular datasets for extreme automatic summarization: WikiHow (Koupaee and Wang, 2018), XSum (Narayan et al., 2018), and SciTLDR (Cachola et al., 2020). WikiHow uses the first introductory sentence of each paragraph of the community-generated instructional how-to articles from *WikiHow.com* as a summary of the paragraph. XSum collects the introductory sentence of news articles from BBC written by professional journalists as a single-sentence summary. SciTLDR uses one-line summaries for academic arXiv papers written by the authors (SciTLDR-Auth) and written by peer reviewers (SciTLDR-Peer). Writing summaries for the same source text but for different purposes likely involves different production strategies (e.g., Hidi and Anderson, 1986, found that certain planning strategies were used for reader-based summaries, but not for writer-based summaries), so we analyzed the two subsets of SciTLDR as two separate datasets in our study. Statistics of the datasets are summarized in Table 1.

### 3.2 Sentence-level Analysis

In order to compress all the important information from the source document into a single-sentence summary, the extreme summary might show a different morphological, syntactic, and/or semantic pattern from the individual source sentences. We measured such compression in extreme summarization by the change in the sentence-level values of each linguistic feature from source to summary. The significance of change indicates which features are consistently manipulated by writers during summarization. We modeled the change of each linguistic feature as a single classification feature in a binary logistic regression model trained to classify sentences as either coming from a source document or from a summary. We also investigated the direction and magnitude of change by comparing the average paired difference of each feature between the source sentences and their associated summary.

#### 3.2.1 Sentence-level features

For each source-summary pair in the datasets, we obtained automatic annotations[1] for both the source document and the summary with features as described below.

[1]Feature data and annotation scripts can be found here: https://github.com/fangcong-yin-2/ling-comp

**Syntactic features**[2]

- **Constituency-parsed tree height** is the constituency tree height of each sentence.

- **Unique part-of-speech tag** is the sentence-level average of the number of unique part-of-speech (POS) tags.

- **Number of modifiers** is the average number of modifiers modifying each token in a sentence. It has been shown to correlate with the cognitive effort involved in creating written text (Ravid and Berman, 2010).

- **Dependency length** is the average dependency length in each sentence.

**Semantic features**

- **Word specificity** represents the semantic distinguishability of noun and verb phrases that represent unique entities, nouns, referents, or events in a given context (Frawley, 1992). The specificity of each noun and verb is measured by the taxonomy depth of the contextualized word sense (obtained by in-context word sense disambiguation) in the WordNet hierarchy (Miller, 1994).[3]

- **Type-token ratio** represents the lexical diversity of a sentence. It is measured by the number of unique word types divided by the total number of tokens in each sentence.

**Morphological features**

Based on previous works in education and children's psychology research (Green et al., 2003; Deacon et al., 2017), the capability to control morphological structures in writing correlates well with the reading comprehension capability for children. Therefore, we also investigated morphological features to quantify the change in sizes of words after summarization in terms of the numbers of semantic word-internal units. Two morphological features are chosen to potentially capture subtle word choices in writing beyond changes in the distribution of different types of words (e.g. the distribution of stop words):

[2]All syntactic features are computed with the Python package *spaCy*.
[3]The part-of-speech tag and the word frequency affects the depth of the synset tree of a word in the WordNet hierarchy. Therefore, we z-scored word specificity of each word by the mean specificity and standard deviation of its part-of-speech tag (rare words were excluded in the calucation of mean).

| Feature | Regression Coefficient (Prediction Accuracy) | | | |
| --- | --- | --- | --- | --- |
| | **WikiHow** | **XSum** | **SciTLDR-Auth** | **SciTLDR-Peer** |
| # Morphemes | -3.842 (0.637) | -2.877 (0.595) | -5.290 (0.705) | -7.404 (0.690) |
| Word Length | -0.192 (0.531) | -0.274 (0.512) | -2.452 (0.768) | -2.218 (0.724) |
| # Unique POS | 0.722 (0.858) | -0.170 (0.573) | 0.333 (0.695) | 0.401(0.720) |
| Tree Height | 0.678 (0.810) | -0.152 (0.593) | 0.194 (0.632) | -0.128 (0.573) |
| Avg # Modifiers | 0.034 (0.498) | -0.299 (0.590) | 0.004 (0.520) | 0.046 (0.493) |
| Avg Dependency Length | 2.369 (0.746) | -0.174 (0.518) | 0.822 (0.604) | 0.756 (0.610) |
| Type-Token Ratio | -22.308 (0.772) | -3.938 (0.547) | -4.945 (0.661) | -0.787 (0.525) |
| Word Specificity | -0.155 (0.554) | -0.500 (0.587) | -0.107 (0.501) | -1.448 (0.597) |

Table 2: Summary of separate logistic regression models using each linguistic feature in isolation to predict whether a text is a source document (label=1) or a summary (label=0). Each model was tested on held-out validation data to obtain prediction accuracy. Prediction accuracies lower than the random baseline (50%) are greyed out.

- **Number of morphemes**[4] is the average number of morphemes in words in each sentence.

- **Word length** is the average number of letters in words in each sentence.

Given that source documents have multiple sentences, we computed grand averages of the sentence-level feature averages over the source documents.

### 3.2.2 Sentence-level Results

Table 2 summarizes logistic regressions using each linguistic feature separately to predict whether a sentence is a summary or from the source.[5] A *positive* coefficient indicates greater compression of a linguistic feature at the sentence level (larger values in the source sentences compared with the summary) while a *negative* regression coefficient indicates expansion. Higher prediction accuracy indicates higher distinguishability of the feature between source sentences and summary and thus greater consistent manipulation by writers during summarization.

In Figure 1, we visualize the change of each feature as the average paired difference between each feature in the source sentences and in the associated summary. We first normalized feature values by min-max normalization to account for the scales of different features (unnormalized raw numerical values can be found in Appendix B). A

positive feature difference indicates a larger value in the source sentences relative to the summary sentence.

In Figure 1, the average number of morphemes and word lengths are *larger* for summaries across datasets of different genres, indicating that summaries tend to be morphologically expanded. Similarly, summaries tend to use more specific words and be lexically more diverse than source sentences. Between the two feature groups, the morphological features more clearly distinguish between summary and source than semantic features as the prediction accuracies are higher in Table 2.

Syntactic features, especially constituency-parsed tree height and the number of unique POS tags, are strong indicators of whether a sentence is a summary or not as suggested by their high prediction accuracies. However, the regression coefficients display genre-dependent variability. XSum summaries tend to be syntactically expanded than the source sentences while WikiHow and SciTLDR summaries are syntactically compressed.

Between SCiTLDR-Auth and SciTLDR-Peer, the change patterns of linguistic features are similar except for syntactic tree height. The similarity implies that when writers create summaries for the same genre of scientific academic paper but for different audiences, they apply similar linguistic compression and expansion strategies.

It is noteworthy that the average number of modifiers is neither an above-random-baseline indicator nor significantly changed from the source to summary, implying that the number of modifiers is not an important factor for extreme summarization.

---

[4]This was computed by the pre-trained morphology annotator *morfessor* (Smit et al., 2014).

[5]We also confirmed the results of our individual regressions using a regression involving joint factors for syntax, semantics, and morphology (see Appendix A).

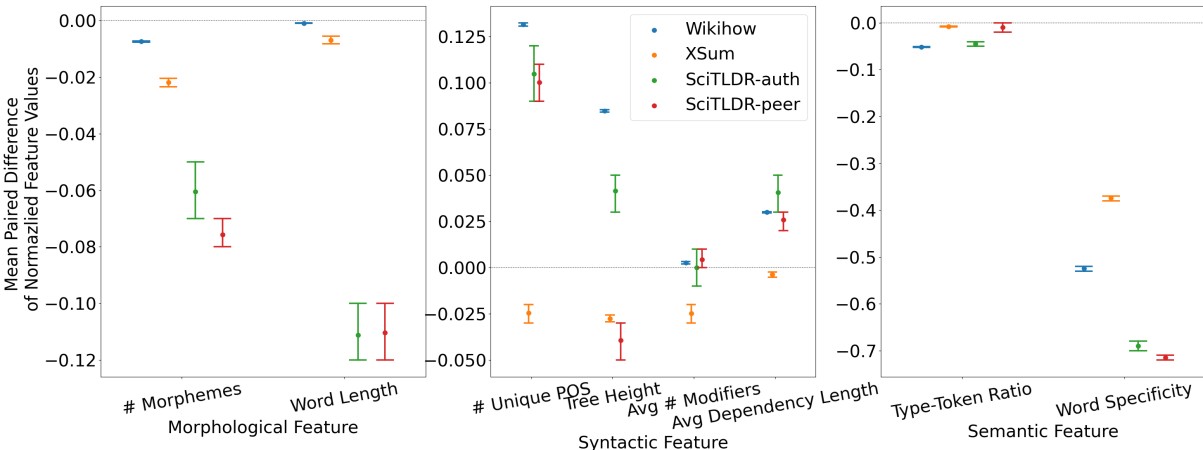

Figure 1: Paired differences of normalized linguistic features between source and summary sentences (source minus summary). A positive value means the feature was compressed during summarization (greater in the source), while a negative value indicates expansion (greater in the summary). Whiskers show 95% level confidence intervals.

This is contrary to findings of previous works that the number of modifiers used in a written text is a good indicator of the writer's capability of processing complex information and handling language (Ravid and Berman, 2010; Durrant and Brenchley, 2022), but it might be grounded in the differences between extreme summary writing and other types of writing (e.g., Ravid and Berman, 2010, examine paragraph-long compositions that describe an event).

## 3.3 Word-level Analysis

We also examined the feature contours within each sentence to investigate whether there are consistent positional patterns in how word-level linguistic features are distributed within each sentence. We generated separate average feature contours for source and summary sentences and qualitatively analyzed the contour patterns. We applied dynamic time warping (DTW) with a barycenter averaging technique (DBA; Petitjean et al., 2011) to align the sentence features and obtain the average contours. DBA approximates the average sequence of a group of sequences by recursively updating the average of aligned sequences to minimize the sum of DTW distances from the average. It has been shown to be a more accurate and informative representation of the average pattern of a group of time series than geometric averaging (Forestier et al., 2017). We first z-scored the features in each sentence and then interpolated each sentence to a length of 100 for standardization. We then applied DTW to align the sequences of each feature and used barycenter averaging to find the average sequence of each feature

in the aligned set.[6] We averaged the DBA outcome over 3 separate runs. As a baseline to represent the positional patterns present in random English sentences, we incorporated the average contours of the test split of the English portions of the English-to-German parallel multi-genre corpus OPUS-100 (Zhang et al., 2020).

### 3.3.1 Word-level features

We analyzed the word-level behavior of two of the features that have shown strong influences in our previous sentence-level analysis.

- **Word specificity** uses the same definition as in the sentence-level analysis, without averaging over each sentence.

- **Dependency length** reflects the psycholinguistic cost of resolving a dependency arc (Gibson, 2000) and has been shown to correlate well with human syntactic processing (Demberg and Keller, 2008; Grodner and Gibson, 2005; Warren and Gibson, 2002; Lewis et al., 2006). The word-level incremental dependency length is computed by summing the length of a terminating dependency along with the number of ongoing unresolved dependencies in a left-to-right parse.[7]

---

[6]We initialized the averaging process using a randomly selected sentence from the data as the suggested by Petitjean et al. (2011). To make sure the initialization is not skewed by the distribution of sentence length, we randomly selected one sentence of each possible sentence length (before interpolation) as an initial average, updated this average with sentences of the same length for 10 iterations using DBA, and finally applied DBA to all the resulting averages of different initializations to obtain a final grand average.

[7]In the original definition of Gibson (2000), dependency

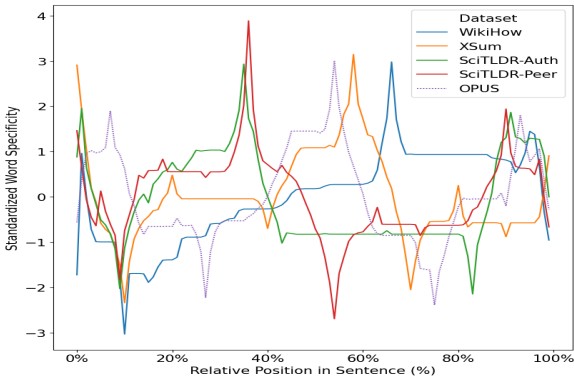

(a) Word specificity of summary sentences

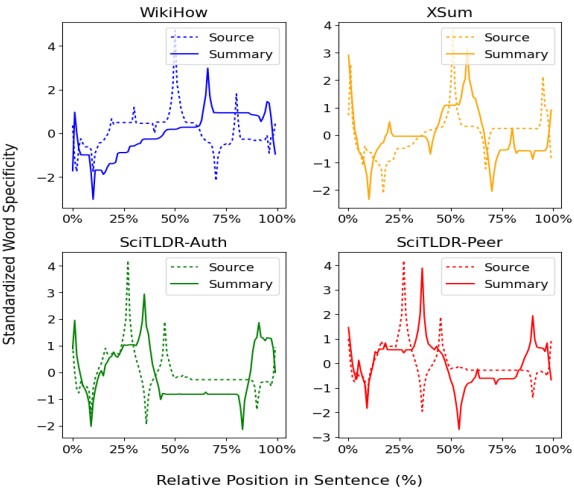

(b) Comparisons of word specificity contours between summary and source sentences

Figure 2: Average contours of word specificity.

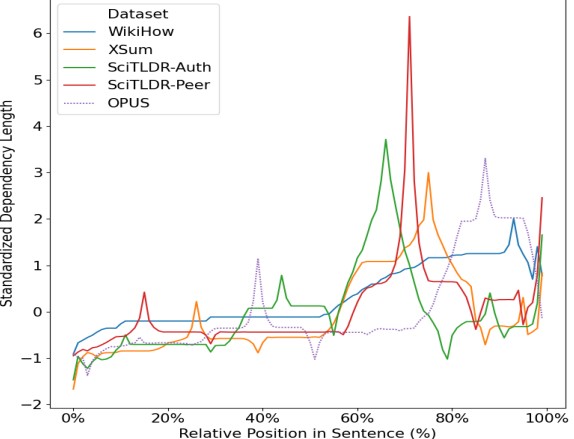

(a) Dependency length of summary sentences

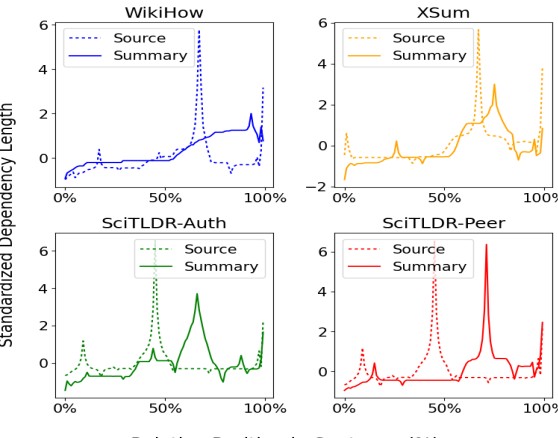

(b) Comparisons of dependency length contours between summary and source sentences

Figure 3: Average contours of dependency length.

### 3.3.2 Word-level Results

Figure 2(a) shows that summary sentences generally start with a relatively high word specificity followed by a drastic dip before in the first $10\%$ of the sentence and reach the peak of word specificity near the $40\%$ location of the sentence. Summaries contours have several different (at some locations, even opposite) high and low locations from the OPUS contour. By comparing the contours between the summary and source sentences in Figure 2(b), we can see that summary contours are mostly the same as those of the source sentences. This indicates that summary sentences might be written to replicate the positional arrangements of specific words in sentences from the source. This is likely a consequence of the fact that specific words exhibit consistent semantic roles through-

---

length is measured using intervening nouns and verbs. In our case, we still measured the dependency lengths of nouns and verbs, but we used *intervening tokens* rather than intervening nouns and verbs only.

out the source document, which is repeated in the summaries.

Figure 3(a) shows that summaries generally have a small spike of dependency length near the beginning of the sentence and a high peak near the end of the sentence (the WikiHow contour is an exception as it does not have a small spike in the beginning; this is because WikiHow summaries are mostly short imperative sentences). Similar patterns can be found in the dependency length contours of OPUS in 3(a) and of the source sentences in 3(b). The similarity indicates that the construction of dependencies in summary and source sentences is similar to that of random sentences in that long dependencies are resolved near the end of sentences.

## 4 Reader Preference Evaluation

Common summarization strategies among writers do not necessarily entail quality: good strategies

of summary writing should help readers better understand the gist of the source, and readers' perceptions of summary texts might be different from the writers' intentions. Therefore, we evaluated the extent to which readers preferred each of the compression patterns described in the previous section.

## 4.1 Study Setup

Because XSum source documents are news articles that require little background knowledge to comprehend and XSum summaries contain richer information from the source given their length, we randomly sampled 20 source articles that have less than 400 words from XSum along with their summaries for our human evaluation.

Using the crowdsourcing platform ProlificAcademic, we recruited 6 native English speakers for each source-summary pair to read the source and summary and rate 7 aspects of summary quality (defined below) on a 1-5 scale. Exact protocols can be found in Appendix C.

**Overall Quality** is the overall quality of the summary.

**Fluency** is the grammaticality and readability of the summary.

**Relevance** indicates a clear focus and no redundant information in the summary.

**Consistency**[8] indicates factuality and faithfulness to the information that appears in the source document.

**Factual Coverage** is the coverage of important facts from the source.

**Novelty** is the use of new words and expressions that have not appeared in the source to paraphrase existing information.

**Abstraction level**[9] is the use of high-level recapitulations of important concepts and ideas.

In each run, raters were presented with source-summary pairs of two different sources in randomized orders. Each rater was paid the equivalent of $12 per hour. Low-quality ratings were filtered based on the correct responses to manually-constructed comprehension questions (2 per source document) and task completion speed (removed if

---

[8]The definitions of fluency, relevance, and consistency are adapted from Fabbri et al. (2021).

[9]Novelty and abstraction level are used to evaluate the abstractiveness of summaries, the degree to which the summary is not copied from the source text yet retains the semantics of the source text (Katwe et al., 2022). Abstractiveness is an important characteristic of extreme summaries and abstractive summaries in general.

< 60s per pair). We collected additional human data to replace the filtered items.

We obtained automatic annotations for linguistic features of the summary only and the paired differences of linguistic features between the source and summary. Features in the summary alone can only be correlated with overall quality, fluency, and abstraction level because these quality aspects could be judged by reading the summary sentence alone while others have to be determined with reference to the source text.

## 4.2 Results

**Intercorrelation of quality aspects** To explore which quality aspects readers emphasize, we fit a linear regression model to predict the overall quality rating using the other quality aspects. We removed any insignificant ($p>0.05$) or strongly multicollinear (variance inflation factor > 10) quality aspect from the regression models. Results show that readers prefer summaries with high novelty ($t=2.39, p=0.019$), relevance ($t=6.604, p<0.01$), and factual coverage ($t=3.178, p<0.01$). They reveal a reader preference for summaries with a clear focus, comprehensive coverage of facts in the source, and new expressions of existing information. These correlations align with the findings of Sanchan (2018) for reader preferences of summaries in the online debate domain: participants prefer summaries that are well-arranged with a concise list of logical points.

**Linguistic features and summary quality.** Values of linguistic features and the rating scores have different scales, and ratings by different raters display a lot of variance. Therefore, we utilized representational similarity analysis (RSA; Kriegestkorte and Bandettini, 2008), which studies second-order rather than first-order relations, to obtain correlations between linguistic features and quality aspects. As a common method used in cognitive science, RSA reveals correlations between measures that operate over different scales. By first correlating each measure with itself across the input stimuli, the scales of the first pass are internally comparable while the outputs are all similarities and therefore are comparable with each other in a subsequent second pass.

We constructed representational similarity matrices (RSMs) for each quality aspect by computing the pairwise cosine similarity score of each stimulus' quality aspect rating with the aspect rat-

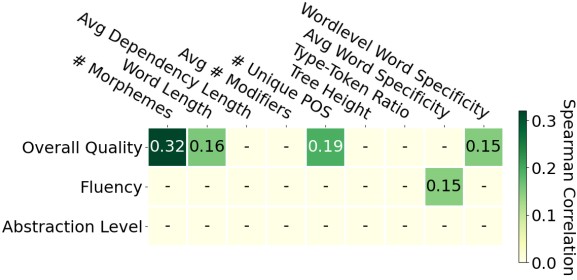

(a) Correlations of features and reader preferences (Summary Only)

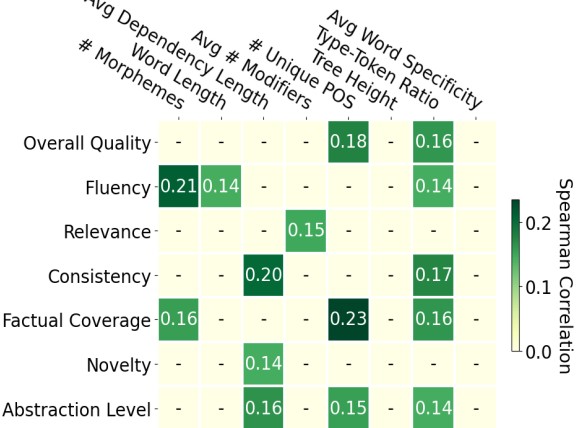

(b) Correlations of changes and reader preferences (Source-Summary Paired Differences)

Figure 4: Significant ($p < 0.05$) RSA correlations between quality ratings and linguistic features. (a) shows that more morphemes and unique POS tags in a summary are correlated with overall quality. (b) shows that changes in type-token ratio and changes in syntactic features are correlated with most quality aspects. Non-significant RSA correlations are represented by "-".

ings of all other stimuli. Similarly, we constructed RSMs for each sentence-level linguistic feature by computing the pairwise Euclidean distance (converted to a similarity score) of that feature across all the stimuli. For word-level linguistic features, DTW distance average is used instead of the Euclidean distance and is converted to the similarity metric. We computed the final second order RSA correlations as the Spearman rank-order correlation between RSMs.

Figure 4(a) summarizes RSA correlations for the linguistic features of the summary alone while Figure 4(b) examines the paired differences of features between the source sentences and summary. Figure 4(a) shows that morphological features and the number of unique POS tags of the summary sentence are strongly correlated with the overall quality. Therefore, both writers and readers consider

these linguistic features important when dealing with extreme summaries. Figure 4(b) indicates that changes in type-token ratio from the source sentences to the summary are correlated with most quality aspects for readers, but this feature did not receive much attention from summary writers as the small average paired differences in Figure 1 indicated. Conversely, we see that changes in word specificity are hardly associated with summary quality by readers while writers emphasize it (as the large average paired differences in Figure 1 indicated). Finally, writers should take note that syntactic features, especially average dependency length and the number of unique POS tags, are correlated with the readers' perceptions of the factual (consistency and factual coverage) and innovative qualities (novelty and abstraction level) of summaries.

## 5 Discussion

This work provides a large-scale computational corpus study towards understanding how humans generally write single-sentence summaries. We performed an analysis of changes in morphological, syntactic, and semantic features and used dynamic time warping with barycenter averaging to analyze the fine-grained positional patterns of linguistic features. Our analyses found that syntactic and morphological changes are helpful when distinguishing whether a sentence is a summary or from a source document. Single-sentence summaries are generally written with morphological expansion, increased specificity of words, and increased lexical diversity compared to the source sentences. At the word level, summaries are written to replicate the positional arrangements of their source sentences while the temporal arrangements of long dependencies in summary sentences are similar to those of random sentences.

We also evaluated the impact of linguistic feature compression on human readers. Using representational similarity analysis over correlations between linguistic features and quality ratings, we showed that morphological features and lexical diversity (type-token ratio) are correlated well with quality ratings. Readers' perceptions of consistency, factual coverage, novelty, and abstraction level of summaries are correlated with changes in syntactic features such as the average dependency length. The results suggest that summary writers and readers both emphasize certain syntactic features, but

type-token ratio and word specificity exhibit a mismatch between writer usage and reader preference. This implies that semantic compression and expansion need to be carefully applied by summary writers to effectively express the main information to readers.

It should be noted that we only indirectly tested the influence of linguistic feature compression on readers through our human study since we constrained our survey to summaries obtained from corpora. In the future, we plan to incorporate direct testing by asking readers to rate summary stimuli that are specifically manipulated with particular compression patterns. Future works might also include investigations of the incremental processing of text production under experimental setups with the help of techniques such as eye tracking and cloze tasks. As motivated by the emerging focus of text summarization in the NLP community, our work can serve as a psycholinguistically-informed resource for improving automatic summarization systems (e.g., explicitly imitating human behaviors during summary writing improves coherence and organization of text summarization; Jaidka et al., 2013; Xiao et al., 2022) and probing model understanding of text generation.

## Limitations

Our corpus analyses only focus on expository texts. Studies have shown that linguistic compressions applied by grade students are substantially different for expository, argumentative, and descriptive texts (Jagaiah and Kearns, 2020). While single-sentence summaries are less common for these genres and thus very limited data exists for large-scale analysis, the observations drawn from our study might be limited to the expository genre. They might also be limited to single-sentence summaries because discourse factors, which are used in general unconstrained summaries, are not considered in our study. However, given that extreme summarization is increasingly common in online settings with lots of real-world applications, we view this work as an important contribution to the understanding of how humans summarize texts.

Regarding human evaluation, the rating data show large variability that might result in insignificant first-order correlations between linguistic features and quality ratings. During multiple pilot runs, we attempted to reduce the variability by increasing the sample size and the number of raters

and filtering out low-quality data points, but we found it difficult to obtain data with high agreement among raters. The variability might be caused by the nature of evaluating the summary quality as the interrater agreements in human evaluations of text summarization are generally low (Iskender et al., 2021).

## Ethics Statement

The human evaluation was conducted under IRB approval. Experimental stimuli were carefully selected from the random samples to not include topics and descriptions of violence and sensitivity, and raters were presented with the option to leave at any time during the study.

Factuality and faithfulness are one of the most important qualities of summaries, and applying strategies to match readers' preferences should not be in trade-off at any time with inaccurate, hallucinated, or distorted summaries of facts. While in-depth studies of the factuality of summaries might need further efforts of fact-checking and de-biasing, we have taken it into account in the study by discussing the quality aspects of consistency, factual coverage, and relevance in detail.

## Acknowledgements

This work is supported by Cornell Engineering Learning Initiatives Undergraduate Research Grant and Wood Excellence Engineering Edu Award. We thank John R. Starr, Maria Antoniak, Jacob Matthews, Debasmita Bhattacharya, and members of the Cornell NLP group and Cornell Computational Psycholinguistics Discussion (C.Psyd) group for their valuable advice. We also appreciate the precious feedback from participants of the Second Workshop on Processing and Evaluating Event Representations (PEER2023) at the University of Rochester and from the reviewers.

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

# A  Multi-factor Logistic Regression

To confirm the results from our individual regressions, we selected one significant feature from each of the feature categories (morphological, syntactic, and semantic) in combination to fit logistic regression models for the same classification task as in our sentence-level analysis. The results in Table 3 confirm our prior observations on the relative ranking of distinguishability among features and the genre-dependent variability for the syntactic feature.

# B  Difference of Unnormalized Linguistic Features

Table 4 includes the difference in unnormalized linguistic features between the source sentences and summary. All the means of paired differences are statistically significant from 0 ($p<0.05$) except for the ones greyed out. Statistical significance is tested with paired t-test if the distribution is approximately normal, and with the Wilcoxon signed-rank test otherwise.

| Dataset | Regression Coefficient | | | Accuracy |
|---|---|---|---|---|
| | # Morphemes | Tree Height | Type-Token Ratio | |
| **WikiHow** | -5.463 | 34.862 | -20.857 | 0.916 |
| **XSum** | -6.247 | -7.190 | -2.550 | 0.657 |
| **SciTLDR-Auth** | -5.579 | 1.472 | -3.714 | 0.779 |
| **SciTLDR-Peer** | -8.344 | -3.213 | -5.766 | 0.795 |

Table 3: Summary of logistic regression models using three significant linguistic features in combination to predict whether a text is a source document or a summary.

| Feature | Dataset | Mean | Median | Q1 | Q3 |
|---|---|---|---|---|---|
| # Morphemes | WikiHow | -0.082 | -0.055 | -0.178 | 0.059 |
| | XSum | -0.040 | -0.025 | -0.126 | 0.063 |
| | SciTLDR-Auth | -0.111 | -0.069 | -0.188 | 0.016 |
| | SciTLDR-Peer | -0.814 | -0.729 | -1.322 | -0.202 |
| Word Length | WikiHow | -0.101 | -0.054 | -0.593 | 0.434 |
| | XSum | -0.055 | -0.018 | -0.419 | 0.338 |
| | SciTLDR-Auth | -0.087 | -0.070 | -0.158 | 0.013 |
| | SciTLDR-Peer | -0.587 | -0.557 | -1.008 | -0.097 |
| # Unique POS | WikiHow | 4.867 | 5.167 | 3.000 | 7.000 |
| | XSum | -0.880 | -0.800 | -3.000 | 1.000 |
| | SciTLDR-Auth | 2.019 | 2.000 | -0.099 | 4.467 |
| | SciTLDR-Peer | 1.747 | 1.856 | 0.000 | 3.744 |
| Tree Height | WikiHow | 3.563 | 3.750 | 2.000 | 5.500 |
| | XSum | -1.007 | -0.800 | -3.200 | 1.250 |
| | SciTLDR-Auth | 1.123 | 1.232 | -0.822 | 3.462 |
| | SciTLDR-Peer | -0.772 | -0.636 | -2.625 | 1.529 |
| Avg # Modifiers | WikiHow | 0.034 | 0.000 | 0.000 | 0.000 |
| | XSum | -0.268 | -0.050 | -0.875 | 0.125 |
| | SciTLDR-Auth | 0.003 | 0.000 | -0.444 | 0.500 |
| | SciTLDR-Peer | 0.040 | 0.000 | -0.500 | 0.564 |
| Avg Dependency Length | WikiHow | 0.497 | 0.532 | 0.126 | 0.902 |
| | XSum | -0.026 | 0.000 | -0.361 | 0.329 |
| | SciTLDR-Auth | 0.178 | 0.236 | -0.171 | 0.622 |
| | SciTLDR-Peer | 0.161 | 0.257 | -0.198 | 0.626 |
| Type-Token Ratio | WikiHow | -0.052 | -0.055 | -0.087 | -0.024 |
| | XSum | -0.008 | -0.012 | -0.048 | 0.028 |
| | SciTLDR-Auth | -0.030 | -0.044 | -0.085 | 0.011 |
| | SciTLDR-Peer | -0.003 | -0.009 | -0.062 | 0.043 |
| Word Specificity | WikiHow | -0.058 | -0.039 | -0.456 | 0.366 |
| | XSum | -0.077 | -0.065 | -0.334 | 0.209 |
| | SciTLDR-Auth | -0.080 | -0.085 | -0.309 | 0.168 |
| | SciTLDR-Peer | -0.126 | -0.103 | -0.324 | 0.106 |

Table 4: Statistics of the paired difference of the unnormalized values of linguistic features between source sentences and summary. Means that are not significantly different from 0 (p>0.05) are greyed out.

## C   Survey Questions of the Human Evaluation

**Overall Quality** What is the overall quality of the summary?

- Very Poor

- Poor

- Barely Acceptable

- Good

- Very Good

**Fluency** How is the fluency of the summary? A fluent summary should be easy to read. It should NOT have capitalization errors or obviously ungrammatical sentences (e.g., fragments, missing components) that make the text difficult to read.

- Very Poor

- Poor

- Barely Acceptable

- Good

- Very Good

**Relevance** How is the relevance of the summary? A relevant summary should have a clear focus and sentences should only contain important information from the source text. It should NOT have irrelevant content or redundant repetition of information from the text.

- Very Poor

- Poor

- Barely Acceptable

- Good

**Very Good**

**Consistency** How is the consistency of the summary? A consistent summary should extract facts from the text and should NOT have information that contradicts the text or does not appear in the text.

- Very Poor

- Poor

- Barely Acceptable

- Good

- Very Good

**Factual Coverage** How is the factual coverage of the summary? A summary with high factual coverage contains all or most of the important information and facts in the source text. A summary with low factual coverage misses some important facts in the source text.

- Very Low

- Low

- Medium

- High

- Very High

**Novelty** How is the novelty of the summary? A novel summary should use new words and phrases that do not appear in the source text to paraphrase the same information. It should NOT be a verbatim copy of the exact words and phrases from the source text.

- Very Poor

- Poor

- Barely Acceptable

- Good

- Very Good

**Abstraction Level** What is the abstraction level of the summary? A highly abstract summary should be a high-level recapitulation of important concepts and ideas in the text. It should NOT be a verbatim copy of technical details from the source text.

- Very Low

- Low

- Medium

- High

- Very High