# OpenReview forum: "Linguistic Compression in Single-Sentence Human-Written Summaries"
_EMNLP/2023/Conference — EMNLP 2023 Findings_

### Official Review · Reviewer_fk6U · 2023-08-04

**Soundness:** 4

**Excitement:**

4: Strong: This paper deepens the understanding of some phenomenon or lowers the barriers to an existing research direction.

**Paper Topic And Main Contributions:**

This paper discusses extreme summarization (one sentence) and compares automatic summarization systems to features of quality human summarization. It further provides human evaluation of automatic summarization and reports the correlations among observed features and human ratings. The novel and interesting thing about this paper is the heavy use of literature on summarization as a human activity and the way in which insights in that literature are turned into features that can be measured in both human and automated summaries.

**Reasons To Accept:**

This paper could improve automatic summarization quality. It is refreshing to see the effort put into consolidating traditional and automated approaches.

**Reasons To Reject:**

I do not see a reason to reject this paper.

**Reproducibility:**

4: Could mostly reproduce the results, but there may be some variation because of sample variance or minor variations in their interpretation of the protocol or method.

**Reviewer Confidence:**

3: Pretty sure, but there's a chance I missed something. Although I have a good feel for this area in general, I did not carefully check the paper's details, e.g., the math, experimental design, or novelty.

---

> ### Author Rebuttal · Authors · 2023-08-29
>
> We would like to thank you for your time and consideration in reading our manuscript. We are really excited about this direction of studying summarization from a human-centered perspective.

---

### Official Review · Reviewer_xvMe · 2023-08-05

**Soundness:** 3

**Excitement:**

3: Ambivalent: It has merits (e.g., it reports state-of-the-art results, the idea is nice), but there are key weaknesses (e.g., it describes incremental work), and it can significantly benefit from another round of revision. However, I won't object to accepting it if my co-reviewers champion it.

**Paper Topic And Main Contributions:**

This paper presents an analysis of human generated single-sentence 'headline' summaries. The analysis compares various aspects of the original source stories and companion summaries. The analyses were conducted on over a million article summary pairs from three data sets from three different genres (instructional texts, news stories and academic/scientific papers). The importance of various features in summarization was computed using a regression model for classifying sentences as belonging to either source/summary buckets. Finally, the summary-features were correlated with reader preferences using feedback from a reader-panel.

**Reasons To Accept:**

Very clear, well motivated problem statement; nicely replicable experiments on reasonably large datasets (which are easily accessible datasets); interesting problem, especially since so many of the use cases for all the deep learning approaches involve summarization (but the actual process is non-transparent). The analysis is useful in understanding how summaries differ from the source documents in terms of their linguistic features and will likely (at least it did for me) raise interesting questions of how we may want to structure systems to explicitly model some of the desired transformations for an improved reader experience.

**Reasons To Reject:**

The analysis while well done yields no surprising new insights. Some of the correlations between reader preferences and linguistic features are difficult to use in building summarization systems since linguistic choices in the summary are constrained primarily by source material. The reader preferences study was weak with "6 native English speakers" rating the summary quality on 7 aspects.

**Reproducibility:**

5: Could easily reproduce the results.

**Reviewer Confidence:**

4: Quite sure. I tried to check the important points carefully. It's unlikely, though conceivable, that I missed something that should affect my ratings.

---

> ### Author Rebuttal · Authors · 2023-08-29
>
> We greatly appreciate your time in reading our manuscript and your helpful suggestions. We would like to address a few of your comments below:
> 1. ”What are some surprising new insights in the analysis?”
> We acknowledge that our results might not be counter-intuitively surprising or state-of-the-art, but our work used a novel combination of computational methods to study linguistic compression patterns and correlations at scale and to shed light on intuitive hypotheses of the summarization process with quantitative evidence. We agree that the summarization process is typically treated as a black box and we hope to provide tools and a starting point for investigating summarization as a human process in more detail though our findings in this paper were largely intuitive.
> 2. “Some of the correlations between reader preferences and linguistic features are hard to use in building summarization systems.”
> Although our main focus in this paper is to better understand the basic science underlying summarization as a cognitive and linguistic activity, we do believe that the features identified as helpful in our work can be potentially helpful for engineering automatic summarization systems for future works. Such features, like the number of unique parts of speech and the type-token ratio, could be easily manipulated in controllable generation settings, though this is beyond the scope of this paper.
> 3. “The strength of the reader preference study is weak given the number of participants.”
> Regarding our human study, we agree with the feedback on its limited scale. However, since our main focus is on understanding linguistic patterns in summarization, we set up the human study mainly to supplement our quantitative corpus findings, especially considering the cost of scaling up human evaluations for our specific setting where each participant needs to do a significant amount of reading and answer a comprehensive set of questions, and to showcase a first step of this novel direction towards understanding summarization as a human activity. We believe this supplementary study is worth incorporating into the main body of our computational work, and we think our results in the human study are interesting to expand upon in future human-focused studies.
> Thanks again for your helpful comments!

---

### Official Review · Reviewer_XKkb · 2023-08-11

**Soundness:** 3

**Excitement:**

3: Ambivalent: It has merits (e.g., it reports state-of-the-art results, the idea is nice), but there are key weaknesses (e.g., it describes incremental work), and it can significantly benefit from another round of revision. However, I won't object to accepting it if my co-reviewers champion it.

**Paper Topic And Main Contributions:**

This paper presents a corpus study investigating the strategy and patterns employed in creating single-sentence human summaries. The findings indicate that single-sentence summaries typically exhibit higher lexical diversity, contain a larger number of morphemes, and feature longer words when compared to the source text.

**Questions For The Authors:**

- You state that extreme summarization would amplify the linguistic compression patters. Could you please elaborate on this?

- Is there a reason you chose this set of morphological features and not things like numbers of stop words, content words, and function words?

- To improve readability, consider placing figures and tables on the same page as their corresponding references. This would eliminate the need for readers to constantly flip back and forth to interpret figures and tables.

- L345: where the claim of correlation between human syntactic processing and dependency length is stated, please provide an appropriate citation.




**Reasons To Accept:**

- The theoretical insights presented in this study can hold significant value in the field of summarization, offering the potential to provide insight to enhance the generation of summaries that align more closely with human preferences.

- Furthermore, this study has the potential to contribute to the establishment of more general reference-independent metrics for evaluating the desirability of AI-generated summaries, as well as for addressing other text generation tasks.


**Reasons To Reject:**

The focus on extreme summarization (single-sentence summaries) limits the scope and applicability of this work. The outcomes of this study might not necessarily extend to more commonly used summarization tasks that involve longer summaries than a single sentence, and it’s not clear how these results may extend to larger span of text as summary.

**Reproducibility:**

4: Could mostly reproduce the results, but there may be some variation because of sample variance or minor variations in their interpretation of the protocol or method.

**Reviewer Confidence:**

3: Pretty sure, but there's a chance I missed something. Although I have a good feel for this area in general, I did not carefully check the paper's details, e.g., the math, experimental design, or novelty.

---

> ### Author Rebuttal · Authors · 2023-08-29
>
> Thank you for taking the time to read our paper and provide helpful feedback and suggestions. We would like to answer your questions about our work as the following.
> 1. “Explain how extreme summarization could amplify linguistic compression patterns”
> Extreme summarization amplifies linguistic compression because it is inherently abstractive and more compressed compared with other forms of summarization. Being abstractive and having such a tight length constraint helps ensure that maximal compression will occur, compared with arbitrary-length summaries, which usually have some form of verbatim copying from the source document. In addition, extreme summary datasets tend to have fewer supplemental materials that are non-essential in the source document (Bommasani and Cardie 2020), providing a purer window on the source compression.
> 2. “Is there a reason you chose this set of morphological features and not things like numbers of stop words, content words, and function words?”
> We specifically chose the two morphological features (the number of morphemes and word lengths) to potentially capture subtle word choices in writing beyond changes in the distribution of different types of words (e.g. stop, function, content words). This assumption comes from education and children’s psychology research that control of morphological structures in writing correlates with capabilities of reading comprehension (e.g. McCutchen et al. 2003 and Deacan et al. 2015). Morphological features are also natural to use to study compressions because they quantify the change of sizes of words in terms of the numbers of semantic word-internal units. We will revise the paper to provide more explanation of this.
> 3. “To improve readability, consider placing figures and tables on the same page as their corresponding references.”
> Thank you for this suggestion. We will take this into consideration in the future revision of our work.
> 4. “Where the claim of correlation between human syntactic processing and dependency length is stated, please provide an appropriate citation.”
> We will provide more citations on the correlation between human sentence processing and dependency length besides the citation of Gibson 2000 in footnote 7. The following is a representative but not exhaustive list of literature that examines such correlation: Demberg and Keller 2008; Grodner and Gibson 2005; Warren and Gibson 2002; Lewis et al. 2006. (We will give the full references in the end)
> 5. “The focus on extreme summarization limits the scope and applicability of this work.”
> We agree that our results on the linguistic compression of extreme summaries may or may not fully generalize to longer summaries, which often involve direct copying of the document. However, our work focuses on summarization as a human activity and linguistic process, which has not been studied thoroughly in the community before. Investigating extreme summaries is a natural first step because it is a natural task with many real-world applications (e.g., headline writing, topic sentence writing, gist distillation) from both the human and machine perspectives and has long been a very influential setup in the automatic summarization community. In addition, most computational linguistics tools work significantly better in terms of reliability and interpretability at the sentence level than at the paragraph/document level. We are not focusing on generalizing to summaries at normal or arbitrary length in this work, but this is indeed a very interesting question that we are interested in pursuing in the future.
> Thanks again for your helpful comments. We hope the above answers can address your questions, and we will include the above clarifications in revisions of the paper!
>
> Reference
> Rishi Bommasani and Claire Cardie. 2020. Intrinsic Evaluation of Summarization Datasets. In Proceedings of the 2020 Conference on Empirical Methods in Natural Language Processing (EMNLP), pages 8075–8096, Online. Association for Computational Linguistics.
> Green, L., McCutchen, D., Schwiebert, C., Quinlan, T., Eva-Wood, A., & Juelis, J. (2003). Morphological Development in Children's Writing. Journal of Educational Psychology, 95(4), 752–761. https://doi.org/10.1037/0022-0663.95.4.752
> Deacon, S. H., Tong, X., and Francis, K. (2017) The relationship of morphological analysis and morphological decoding to reading comprehension. Journal of Research in Reading, 40: 1–16. doi: 10.1111/1467-9817.12056.
> Demberg V, Keller F. Data from eye-tracking corpora as evidence for theories of syntactic processing complexity. Cognition. 2008 Nov;109(2):193-210. doi: 10.1016/j.cognition.2008.07.008. Epub 2008 Oct 18. PMID: 18930455.
> Grodner, D.J. and Gibson, E.A. (2005) Consequences of the serial nature of linguistic input for sentential complexity. Cogn. Sci. 29, 261–290
> Warren, T. and Gibson, E. (2002) The influence of referential processing on sentence complexity. Cognition 85, 79–112
> Lewis RL, Vasishth S, Van Dyke JA. Computational principles of working memory in sentence comprehension. Trends Cogn Sci. 2006 Oct;10(10):447-54. doi: 10.1016/j.tics.2006.08.007. Epub 2006 Sep 1. PMID: 16949330; PMCID: PMC2239011.

---

### Meta-Review · Area_Chair_nPi7 · 2023-09-19

**Recommendation:** 3

**Metareview:**

The results are sound and convincing but their contribution to linguistics and/or cognition are not immediately perceivable.

---

### Decision · Program_Chairs · 2023-10-07

**Decision:**

Accept-Findings

**Comment:**

The results are sound and convincing but their contribution to linguistics and/or cognition are not immediately perceivable.